# Strategies for Improving Firefighter Health On-Shift: A Review

**DOI:** 10.3390/jfmk9020105

**Published:** 2024-06-15

**Authors:** Kealey J. Wohlgemuth, Michael J. Conner, Grant M. Tinsley, Ty B. Palmer, Jacob A. Mota

**Affiliations:** 1Neuromuscular and Occupational Performance Laboratory, Texas Tech University, Lubbock, TX 79409, USA; kealey.wohlgemuth@ttu.edu; 2Front Line Mobile Health, Georgetown, TX 76048, USA; mike@frontlinemobilehealth.com; 3Energy Balance and Body Composition Laboratory, Texas Tech University, Lubbock, TX 79409, USA; grant.tinsley@ttu.edu; 4Muscular Assessment Laboratory, Texas Tech University, Lubbock, TX 79409, USA; ty.palmer@ttu.edu

**Keywords:** fire service, fatal injuries, non-fatal injuries, training, nutrition, tactical athlete, occupational performance, first responder

## Abstract

The fire service suffers from high rates of cardiovascular disease and poor overall health, and firefighters often suffer fatal and non-fatal injuries while on the job. Most fatal injuries result from sudden cardiac death, while non-fatal injuries are to the musculoskeletal system. Previous works suggest a mechanistic link between several health and performance variables and injury risk. In addition, studies have suggested physical activity and nutrition can improve overall health and occupational performance. This review offers practical applications for exercise via feasible training modalities as well as nutritional recommendations that can positively impact performance on the job. Time-efficient training modalities like high-intensity interval training and feasible modalities such as resistance training offer numerous benefits for firefighters. Also, modifying and supplementing the diet and can be advantageous for health and body composition in the fire service. Firefighters have various schedules, making it difficult for planned exercise and eating while on shift. The practical training and nutritional aspects discussed in this review can be implemented on-shift to improve the overall health and performance in firefighters.

## 1. Introduction

The firefighting occupation is physically demanding due to the numerous responsibilities of the job, shift-style work schedules, and limited ability for recovery. In a 2022 report, there were 1,041,200 career and volunteer firefighters in the U.S. with a 2% increase in career firefighters from the preceding year [1]. In addition, there were 29,452 fire departments; where 18% were career departments that are responsible for protecting 70% of the U.S. population [1]. The demands of the job often put firefighters at a high risk for illness and injury. Recent work shows 60,750 firefighters were injured in the line of duty in 2021 [2]. The estimated cost of firefighter injuries is between $2.8–7.8 billion per year [3]. The most commonly reported causes of injury stem from overexertion or stress on the job and falls, slips, and trips [2]. 

Firefighters often are deemed to have poor overall health as the fire service suffers from obesity, below-standard performance metrics, and high rates of cardiovascular disease [4,5,6,7] (Figure 1). A previous study suggested firefighters may present with subclinical cardiac dysfunction, which is associated with obesity, hypertension, poor metabolic health, and low aerobic capacity [6]. Sedentary workdays when not responding to emergencies plus unhealthy dietary habits hinder health in the fire service. It may be difficult for some fire departments to monitor exercise and provide nutritional information to employees. Also, nutrition within the fire service is unique due to several factors such as communal eating and unnatural mealtimes. While firefighters often have poor performance and health metrics, they are still called upon to perform maximally and efficiently when responding to emergencies. Firefighters have recently been referred to as “tactical athletes” or “occupational athletes” because their occupation requires them to run, lift, and perform high-intensity tasks to the best of their ability. Exercise and health promotion is needed in the fire service through training modalities that are advantageous to firefighters and fit various work schedules. It is important that the training programs improve cardiometabolic and musculoskeletal health within firefighters. Also, feasible training modalities (e.g., minimal equipment and short duration) are needed to fit various shift schedules of various fire departments. In addition, the fire service should provide information regarding nutritional strategies to promote health and performance in firefighters. The purpose of this review is to synthesize the literature involving training and nutrition that can be adopted in the fire service to promote overall health and occupational performance in all firefighters. A secondary purpose of the review is to discuss practical strategies to improve training and nutrition within the fire service.

## 2. Methods

A review of the literature regarding shiftwork and exercise related to the fire service was completed. Also, previous works discussing diet and nutrition within firefighters were reviewed. A search of the PubMed database in September 2023 was conducted using phrases and terms such as, “firefighter training”, “firefighter exercise”, “firefighter performance”, “firefighter injuries”, as well as “firefighter body composition”, “firefighter cardiovascular”, and “firefighter fasting”. In addition, combined terms such as “fire service and nutrition”, “fire service and exercise”, and “firefighter training or firefighter exercise” were searched. Additionally, papers examining adaptations from high-intensity interval training and resistance training were searched due to the lack of studies investigating exercise and nutritional strategies in the fire service. The literature regarding nutrition and dietary supplements in tactical populations was also included. 

### 2.1. Training

#### 2.1.1. Training On-Shift

Shiftwork within the fire service varies depending upon departments; however, the most common schedule is a 24 h shift (e.g., 6:30 am–6:30 am) [8] paired with 48 h off duty [9]. During the time on-shift, firefighters respond to calls for various emergencies such as structure fires, motor vehicle collisions, and aiding emergency medical services. During the 24 h shift, the National Fire Protection Association (NFPA) Standard 1583 states that fire departments should provide the means for regular exercise and firefighters assigned to a shift will be allowed time to exercise while on-shift [10]. The Standard 1583 also lists seven components that a fire department’s exercise program must contain: an educational program, individualized exercise programs, warm-up and cool-down guidelines, metabolic conditioning (aerobic and anaerobic), muscular resistance, flexibility and mobility, and an emphasis on the back, knees, and shoulders [10]. Implementing training on-shift can vary and may be difficult to accomplish depending on call volume and emergency response obligations.

#### 2.1.2. Types of Training

Interval training, specifically, high-intensity interval training (HIIT) is a modality which improves both the cardiovascular and musculoskeletal systems [11,12]. Interval training consists of repeated bouts of high-intensity exercise paired with bouts of recovery [13] and is often deemed as a time-efficient exercise modality [14,15]. A unique aspect of HIIT is that it can be completed using cycle- or running-based training [16,17]. The little time involved and enjoyment of partaking in HIIT may suggest that it is an appropriate modality for previously sedentary individuals [18]. In addition to HIIT, resistance training is another advantageous exercise modality. Resistance training has been shown to improve functional movement, performance, strength, and cardiometabolic health [19,20]. Both training modalities (HIIT and resistance training) fit within the scope of the previously mentioned NFPA 1583. Aerobic training or endurance training may not be the most feasible for improving the cardiovascular and musculoskeletal systems while on-shift, due to the significant time requirements of this exercise modality. Also, endurance training may not meet the requirements of NFPA 1583 such as metabolic conditioning and muscular resistance. Potential benefits of HIIT and resistance training are explained in Figure 2. 

#### 2.1.3. Adaptations to HIIT

High-intensity interval training improves aerobic and anaerobic systems due to the repeated bouts of high-intensity work followed by periods of recovery [13]. Previous work has shown HIIT to be effective in normal weight [21] and obese [12,22] groups, young [23] and old [24] individuals, and untrained [12] and trained athletes [17]. The frequency of HIIT sessions varies across the literature; however, adaptations can be seen in relativity short time periods (i.e., 3 weeks–8 weeks). 

In just three weeks, cycle-based HIIT was able to increase muscle size (e.g., vastus lateralis cross sectional area) by 14% in overweight and obese males and females [25]. Similarly, a cycle-based HIIT intervention was able to increase the volume of the quadriceps after just three sessions, and quadricep volume increased (+11.82%) from pre- to post-testing following a six-week intervention [26]. These increases in muscle size are likely due to muscle hypertrophy [27]. In normal weight individuals, eight weeks of HIIT improved aerobic capacity and jump performance (i.e., squat jump and countermovement jump) [21]. In overweight and obese groups, eight weeks (2x/week training) of cycle-based HIIT improved muscle size, muscle quality, aerobic capacity, body composition, and cardiometabolic health [12,22,28]. Hirsch et al. [28] also indicated that an eight-week HIIT intervention altered resting substrate metabolism by increasing fat oxidation, with previous work demonstrating that HIIT has been effective at reducing fat [29]. In a study implementing running HIIT three times a week, results showed improvements in aerobic capacity (19.6%) in overweight and obese males [30]. Specific to tactical athletes, Gripp et al. [12] compared eight weeks of moderate-intensity interval training (MIT) to eight weeks of HIIT in overweight or obese police officers. The participants completed a running program once a day for three days per week. The results showed that aerobic capacity, body composition, and metabolic health had greater improvements due to the HIIT than MIT. 

#### 2.1.4. Adaptations to Resistance Training

The goal of resistance training is to promote muscular performance and muscle hypertrophy by stressing the muscle metabolically and putting the muscle under tension [31]. A common repetition range for achieving muscle hypertrophy is six to twelve repetitions with a rest period of thirty to ninety seconds between each set [31]. Resistance training improves muscular strength by both neural and hypertrophic factors which lead to an increase in force production [32,33]. In addition, resistance training increases muscle size [34,35] and muscle quality [36]. Collectively, these studies indicate resistance training can improve muscle function through increasing size, strength, and quality. 

Aagaard et al. [37] reported increases in muscle size (i.e., cross sectional area and volume) and individual muscle fiber size after 14 weeks of lower-extremity resistance training (e.g., hack squats, leg press, knee extension, hamstring curls and calf raises). Specifically, the authors noticed an increase in pennation angle, which contributes to an increased in force-producing quality of the muscle [37]. However, the neural adaptations to resistance training, particularly regarding motor unit firing rate are inconsistent in the literature [38]. Del Vecchio et al. [39] reported that four weeks of resistance training of the ankle muscles (dorsiflexors) was able to increase motor unit discharge rate as well as decrease the recruitment threshold force of motor units. A systematic review and meta-analysis stated that strength training increases corticospinal excitability [40]. Resistance training also improves body composition, particularly through reducing body weight, body fat percentage (BF%), and fat mass, while increasing lean mass in overweight and obese individuals [41,42]. More importantly, a recent review indicates resistance training decreases visceral fat [42], which is important due to the association of higher visceral fat with poor cardiometabolic outcomes [43]. Lastly, resistance training may improve other aspects of cardiometabolic health such as regulating hypertension [44]. 

### 2.2. Nutrition 

#### 2.2.1. Protein Consumption

Nutrition is important to overall health and performance; however, consuming complete meals and normal mealtimes can be difficult for firefighters on-shift. While all macronutrients (i.e., fat, carbohydrate, and protein) should be consumed in the diet to create well-balanced meals, recent work suggests increased protein consumption may be beneficial for active populations. Examples of high-quality proteins that can be consumed are chicken, fish, eggs, dairy, and lean beef, among others [45]. The current Recommended Dietary Allowance (RDA) for protein suggests adults (≥18 years) should consume 0.8 g per kilogram of bodyweight a day to avoid nitrogen loss from the body [46]. Recent work has highlighted that adults benefit from consuming more protein than is stated by the RDA [46,47]. Protein consumption may help body weight regulation as protein has been known to increase satiety compared to fat and carbohydrate, increase thermogenesis due to the energy needed to metabolize amino acids, and improve blood lipid profiles [47,48,49,50]. The International Society of Sports Nutrition (ISSN) position stand on exercise and protein suggests for building or maintaining lean mass in exercising individuals, daily protein intake should be 1.4–2.0 g per kilogram of body weight per day, a recommendation that is typically consistent with the Acceptable Macronutrient Distribution Range of 10 to 35% of total energy from protein [51]. The consumption of adequate amounts of high-quality protein throughout the day, specifically ~30 g of protein per meal is important for maximally stimulating muscle protein synthesis [47]. Additionally, consuming adequate protein with carbohydrates has been shown to decrease muscle damage and increase muscle glycogen stores [52]. During recovery, consuming higher amounts of protein may also be advantageous as protein helps tissue healing and maintaining lean mass [53]. Philips [54] explained that military personnel benefit from consuming higher amounts of protein than recommended due to both the mental and metabolic stress their bodies endure. Although the aforementioned recommendations are for military personnel, firefighters may also see advantages when consuming more protein as they still undergo physiological and mental stress resulting from the physical demands of the occupation. 

#### 2.2.2. Caloric Restriction Modalities 

Previous work has evaluated the role of shiftwork on eating habits in tactical occupations [55,56] due to studies indicating an increased risk of cardiovascular disease [57] and obesity [58] in shift workers. Several nutritional strategies have been proposed to combat the ill effects of shiftwork. While there have been contradicting claims about intermittent fasting, there have been recent works suggesting fasting is a viable modality of manipulating eating behavior which can improve health and body composition [59,60,61]. Intermittent fasting may be an alternative modality of improving body composition instead of traditional forms of caloric restriction [59]. One subtype of intermittent fasting, termed time-restricted eating, may be particularly applicable to shiftwork due to its potential influence on circadian rhythms [62]. Time-restricted eating restricts energy consumption to a certain period of time (e.g., 6–12 h) each day [56] and has been shown to reduce body mass, fat mass, and visceral fat in obese males and females [63]. A previous study [56] completed in firefighters evaluated time-restricted eating during 24 h shifts where firefighters were randomized into two groups: 12 weeks of 10 h time-restricted eating or a standard-of-care group. Results showed that firefighters allocated in the time-restricted eating group had improved quality of life and cardiometabolic health (i.e., decreased blood pressure and improved blood lipid profiles) as well as decreased body weight as a result of the intervention [56]. Gonzalez et al. [64] implemented a 7-week time-restricted eating (14 h fast, 10 h heating) program in firefighters. The results showed that time-restricted eating improved ventilatory threshold and decreased aerobic capacity (VO_2_max) but did not hinder muscular performance variables (strength and endurance) [64]. Studies evaluating caloric restriction interventions in tactical populations are scarce; however, previous time-restricted eating studies [56,65] highlight the feasibility of modifying eating behaviors within the fire service and potential cardiometabolic benefits. 

#### 2.2.3. Dietary Supplements and Ergogenic Aids

It is important to consume whole foods and intake all macronutrients in the diet. However, the literature shows that dietary supplements and ergogenic aids may be able to improve athletic performance [66], enhance cognitive function [67], and help meet caloric needs for males and females [68], which in turn could improve occupational performance in firefighters. In addition, it may be difficult for on-duty firefighters to have time to prepare well-balanced meals; therefore, dietary supplements may be beneficial. However, firefighter performance after supplement or ergogenic aid consumption is understudied [69]. Potential benefits and recommendations for dietary supplements and ergogenic aids are listed in Table 1. 

As mentioned previously, consuming adequate protein is important for active individuals. It may be difficult to consume whole foods and high-quality proteins to meet caloric needs while on duty; therefore, protein supplementation could be advantageous for firefighters. Traditionally, most protein supplements contained milk (e.g., whey and casein) or egg proteins [45], although there is now an increased availability of plant-based proteins (e.g., soy and pea). It has been suggested that whey protein isolate is one of the purest sources of protein (>90% protein concentration, removed fat, removed lactose) [70]. Protein consumption is recommended at 20–30 g boluses to maximally stimulate muscle protein synthesis [51,71,72]. Many of the benefits of protein supplementation may be attributable to their essential amino acid (EAA) content. EAAs should be consumed within daily protein intake or in their free form to help with skeletal muscle protein synthesis [52]. For EAA supplements, it is recommended that 10 g of EAAs are consumed; however, this quantity of EAAs can also be obtained by consuming ~25–30 g of high-quality protein, indicating that supplementation with free-form EAA is not essential for those consuming adequate protein [45]. Particularly, when paired with resistance training, EAAs may help promote muscle protein synthesis and aid in recovery [73,74]. Antonio et al. [75] reported that a 6-week EAA supplementation (18.3 g a day) regime improved aerobic muscular endurance (i.e., time to exhaustion). Firefighters may consider protein supplementation to help meet their protein needs. 

A popular ergogenic aid, caffeine, works upon adenosine receptors and the central nervous system [76]. Caffeine has been noted to enhance alertness during the day and at night [77] and improve athletic performance in some studies [78,79]. Specifically, caffeine has improved anaerobic power in previous work [80] and increased muscle strength [81]. The recommendation for caffeine dosage is 3–6 milligrams per kilogram of body weight [82]. It is important to note that in a recent study [83], consuming caffeine resulted in an increased coagulation response in firefighters completing drills. Caffeine’s ability to improve anaerobic performance as well as improve alertness could be advantageous for firefighters who are physically active while on-shift. Also, firefighters often complete tasks which demand alertness at various times of day, so caffeine may be beneficial.

Recent studies have evaluated the supplement creatine monohydrate and have indicated numerous benefits for overall health and performance in both sexes [84,85,86]. Particularly, creatine is beneficial for high-intensity exercise performance [87,88] as it increases phosphocreatine levels within skeletal muscle [89]. In addition, creatine has shown to be neuroprotective regarding traumatic brain injuries, aid in recovery from brain injuries [90], and helps with overall cognitive functioning [67]. Some studies suggest that creatine may help reduce the effects of sleep deprivation on cognitive function [66]. The most common recommendation for creatine monohydrate supplementation is consuming 5 g a day—or more specifically, 0.3 g per kilogram of body weight—four times a day for five to seven days for loading [89], with maintenance noted as consuming three to five grams daily [86]. Adding creatine monohydrate to the diet may increase training adaptations as well as cognitive functioning in firefighters. 

### 2.3. Impacting Firefighter Health

#### 2.3.1. Reducing Obesity

It is well documented that firefighters are often overweight or obese (body mass index [BMI] ≥ 25 kg/m^2^), and the prevalence of obesity is greater in the fire service when compared to U.S. citizens [91]. One study indicated the majority of recruits are overweight or obese as well [92]. In addition, each year of service is associated with an increase in body mass (0.42 kg), fat mass (0.25 kg), BMI (0.13 kg/m^2^), and BF% (0.18%) [93]. Another study reported an increase in obesity in a 5-year period (~35 to 40%) for firefighters [7]. Obesity is associated with cardiovascular disease and metabolic disease [94], which is important to consider as cardiac-related deaths are the leading cause of line-of-duty deaths in the fire service in 2022 [95]. Poor body composition also impacts occupational performance as firefighters with higher BF% had poorer balance [96], and those with a higher BMI had poorer functional performance [97]. Obesity may be improved using a multi-faceted approach, including behavior change, nutritional considerations, and exercise interventions such as HIIT [98]. For example, Chin et al. [30] implemented HIIT (12 sets x 1 min work at 90% heart rate reserve [HRR]: 11 sets x 1 min recovery at 70% HRR) in three exercise groups (3x week, 2x week, and 1x week) and indicated decreases in body fat and fat mass in all exercise groups compared to the control group. Modifying diets may also improve body composition as previous work has shown firefighters consuming more protein tend to have lower fat mass and BF% than those consuming lower protein quantities [99]. It is also known that supplementation with whey protein in addition to resistance exercise can be beneficial for body composition [100]. Reducing obesity rates and improving body composition in the fire service may occur by implementing exercise programs and diet modifications while on-shift.

#### 2.3.2. Lowering Cardiovascular Disease Risk

Previous reports have highlighted poor cardiovascular health within the fire service [6,101,102], and studies suggest that poor health may be associated with an increased risk of cardiac-related events [103,104]. Smith et al. [6] reported that 63% of firefighters (*n* = 967) indicated subclinical cardiac dysfunction which the authors note was related to obesity, hypertension, and poor aerobic capacities. According to an NFPA report, 54% of firefighter deaths were due to overexertion, stress, or medical reasons in one year, with 24 being cardiac-related deaths [105]. Previous work has noted that fire suppression tasks are correlated to line-of-duty deaths; however, recent data may suggest that other, non-fire suppression tasks may be related to sudden cardiac death [106]. Recently, there has been an emphasis on understanding how to improve cardiovascular health and lower the rates of cardiovascular disease in the fire service, specifically through exercise interventions [5]. The National Fire Protection Association Standard on Comprehensive Occupational Medical Program for Fire Departments recently updated recommendations to require firefighters to have aerobic capacities equal to the 35th percentile, based on age and biological sex, subject to duty restrictions [107]. Previous reports have indicated that firefighters are not meeting these aerobic capacity standards [5]. Further, it is well documented that exercise interventions (HIIT and resistance training) can improve aerobic capacity [28] and cardiometabolic health [108,109]. High-intensity interval training (3x for 8 weeks) improved aerobic capacity in overweight and obese participants (+5.27%) and their normal weight counterparts (+2.88%) [21]. In addition, Ouerghi et al. [21] indicated total cholesterol (−11.8%), LDL cholesterol (−11.9%), and triglycerides (−21.3%) also decreased in obese and overweight participants. It has also been reported that various HIIT modalities improve vasculature function [11,108]. Lastly, authors who conducted a randomized controlled trial in males concluded that resistance training paired with protein supplementation is beneficial for health [110]. Taken together, exercise training is beneficial for lowering cardiovascular disease risk as well as improving aerobic capacities, which are both are needed within the fire service. When paired with nutritional strategies like consuming adequate protein, a greater impact may be felt on firefighter health.

#### 2.3.3. Decreasing Musculoskeletal Injuries

In addition to high rates of fatal injuries in the fire service, many firefighters also suffer from non-fatal injuries [111,112]. These injuries are often to the musculoskeletal system and in the form of strains and sprains [95,112,113]. Many injuries are the results of slips, trips, and falls [102,114]. In addition, previous works have noted the most common injury sites to be the lower extremities (i.e., the ankles and knees) and the back [111,112]. Studies have shown mechanistic links between neuromuscular variables and injury risk. For balance specifically, Punakallio et al. [115] reported that poor functional balance was related to slip and fall risk, while older firefighters had longer slips. In a similar study assessing functional balance, lower-extremity strength and poor body composition (i.e., higher BF%) were correlated to lower performance on the balance assessment [96]. Muscle quality may also be important in balance performance and fall risk [116]. Lastly, Marciniak et al. [117] indicated that in firefighter recruits, better balance performance was related to BMI, lower-extremity strength, and movement efficiency. Taken together, these findings suggest neuromuscular variables are related to balance performance. While balance is related to fall risk, functional movement is also an important variable related to risk of musculoskeletal injury [118]. Cornell et al. [97] also indicated significant relationships between BMI and functional movement as well as fat-free mass and functional movement in firefighter recruits. Strength imbalances in the lower extremities, particularly the hamstrings to quadriceps strength ratio may allow insight into risk for knee injuries [119,120]. While injuries are noted most often in the lower extremities due to slips, trips, and falls, the trunk is a common injury site in the fire service [121]. Core strength has been noted to help lower-back strength [122]. In an intervention to improve core strength in firefighters, the loss of time at work due to injury was reduced by 62%, and overall injuries were decreased by 42% in a 1-year period in firefighters who participated [123]. Similarly, firefighters participated in a 24-week (2x/week) trunk exercise program, which resulted in 21% greater core muscular endurance and 12% greater back muscular endurance [121]. Further, the risk of musculoskeletal injuries in the fire service is high due to numerous causes which mostly consist of low muscular strength and endurance, poor balance, and inadequate functional movement. Implementing exercise interventions could improve muscular strength, in turn benefitting balance and functional movement in the fire service. 

#### 2.3.4. Improving Occupational Performance

Improving overall health and decreasing injury risk factors could improve occupational performance in firefighters (Figure 3). For instance, performance (i.e., time to task completion) has been assessed during stair climb, which is an occupationally relevant task. Kleinberg et al. [124], reported larger muscle cross sectional area and lower echo intensity of the vastus lateralis and rectus femoris were associated with better stair-climb performance in firefighters. Similarly, Ryan et al. [125] investigated neuromuscular factors impacting stair-climb performance. The results suggest a quicker stair-climb time is related to greater lower-extremity power as well as better body composition (lower BF%) in firefighters [125]. 

In addition, many firefighters complete bouts of simulated occupational tasks during training. A common assessment is an eight-task assessment, most commonly known as the Candidate Physical Abilities Test (CPAT). Previous work has evaluated performance metrics impacting CPAT completion [126,127]. Williams-Bell et al. [127] suggested the CPAT encompasses both aerobic and anaerobic systems with aerobic capacity, heart rate, and anerobic energy demands being high. Sheaff et al. [126] also reported aerobic and anaerobic systems to be important to CPAT performance. In this study [126], performance metrics were compared between firefighters who passed and failed the CPAT. Power determined from a Wingate cycle-based test, aerobic capacity determined via a treadmill test, and upper-body strength were related to passing the CPAT [126]. With previous knowledge of resistance training increasing muscle size and quality, plus HIIT impacting both aerobic and anaerobic systems, both exercise modalities may improve occupational performance. 

### 2.4. Implementation 

Although it is important to note that the time dedicated for training on-shift may vary with each shift depending upon call volume, there is mandatory time built into the shift for exercise. It is important to note that exercise time is available, but performing exercise is optional. Feasible and time-efficient exercise modalities such as HIIT and resistance training would be advantageous to firefighters and easily completed on-shift. Common exercise equipment in fire departments includes treadmills and exercise bikes which can be utilized for HIIT. If a department does not have gym equipment and is using a local gym for access, a treadmill or exercise bike should be accessible. Prescription of HIIT is usually training for a short period of time (“work”) dependent upon heart rate (HR) or rate of perceived exertion (RPE) and then recovering at a lower intensity (“rest”) (i.e., HR or RPE). For example, while “working”, firefighters should be within 80–95% of their maximum HR, but while “resting”, HR should be 40–50% of the maximum [128]. Common ratios are 1:1 for work:rest, meaning 2 min of “work” followed by 2 min of “rest”. Higher-intensity “work” should be followed with longer “rest” (e.g., 30 s sprint:4 min walk) [128].

In addition, most fire departments contain resistance training equipment such as barbells, plates, dumbbells, and kettlebells. If the fire department does not have the means for resistance training equipment, a local gym or fitness center that is being used should have equipment available for resistance training. As common injury sites are the lower extremities and lower back, firefighters should focus improving muscular strength and endurance of the quadriceps, hamstrings, core, and lower back through common lifts such as deadlift, back squat, and front squat. A common prescription for muscular strength is 1–5 repetitions per set at 80–100% of 1-repetition maximum (1RM), muscular hypertrophy is 8–12 repetitions per set at 60–80% 1RM, and muscular endurance can occur following 15+ repetitions per set with an intensity below 60% of 1RM [129]. It may be beneficial for firefighters to consult the departmental strength and conditioning professional, Peer Fitness Trainer, wellness coach, or a trainer at a local outsourced gym for individualized training programs. 

Modifying caloric intake and supplementing the diet may have several benefits for firefighters while on-shift due to the unknown schedule. Meeting the protein recommendations (1.4–2.0 g/kg/day) [45] would be most beneficial for increasing satiety and recovery while on-shift. Further, daily protein consumption for a 50 kg firefighter would be 70–100 g, while for a 100 kg firefighter, it would be 140–200 g. Keeping in mind that ~30 g boluses should be consumed, a 50 kg firefighter should have ~3 boluses a day, while the 100 kg firefighter should have ~5–6 boluses a day. The 50 kg firefighter may only need to consume adequate protein at normal mealtimes (breakfast, lunch, and dinner) or when the schedule allows throughout a shift. Contrarily, the 100 kg firefighter may need protein supplementation to help meet their caloric needs throughout a shift. An example of on-shift recommendations can be found in Figure 4. Following training or calls, consuming whey protein may be important in aiding with muscle recovery. For instance, if a call is particularly strenuous, firefighters may want to consume a whey protein shake (~30 g protein) to help in skeletal muscle protein synthesis and recovery following activity. In addition, if trying to meet training or body composition goals, creatine monohydrate may be beneficial. For example, if a firefighter is routinely training and trying to improve body composition, consuming creatine monohydrate can increase lean mass and muscular strength. A helpful resource for nutritional recommendations specific to the fire service may be the recently published position stand for tactical athletes from the International Society of Sports Nutrition [69]. 

## 3. Conclusions

Poor health, below-standard performance metrics and unhealthy diets plague the fire service, playing a large role in fatal and non-fatal injury rates in firefighters. Recent calls to action have been established for health promotion within fire departments. 

This current work attempted to summarize the literature and offered feasible and effective strategies for firefighters to adopt while on-shift. Secondly, this review highlighted implementation strategies for training as well as practical nutrition recommendations for firefighters to incorporate on-shift. The research regarding exercise and dietary interventions in firefighters is limited; however, results from a few studies show benefits for firefighter health, wellness, and occupational performance (Figure 5). 

## Figures and Tables

**Figure 1 jfmk-09-00105-f001:**
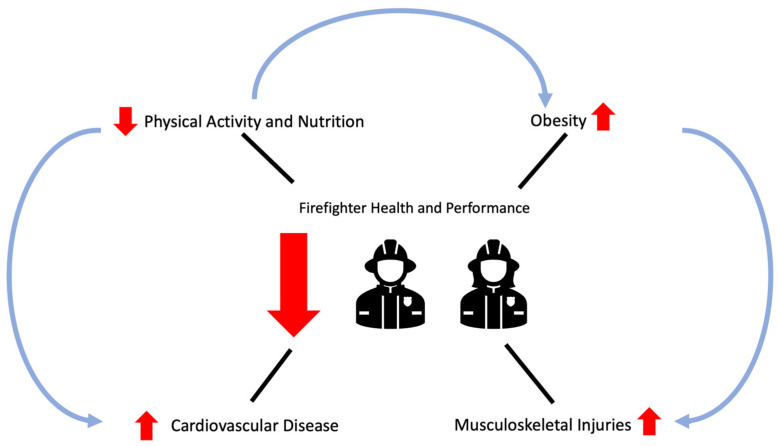
Mechanisms leading to decreased firefighter health and performance are low levels of physical activity, poor nutrition, obesity, high rates of cardiovascular disease, and injuries on the job. Improving physical activity and nutrition can decrease obesity and lower cardiovascular disease risk, in turn improving performance. Poor body composition is a risk factor for musculoskeletal injury. By improving body composition, occupational performance and health may improve.

**Figure 2 jfmk-09-00105-f002:**
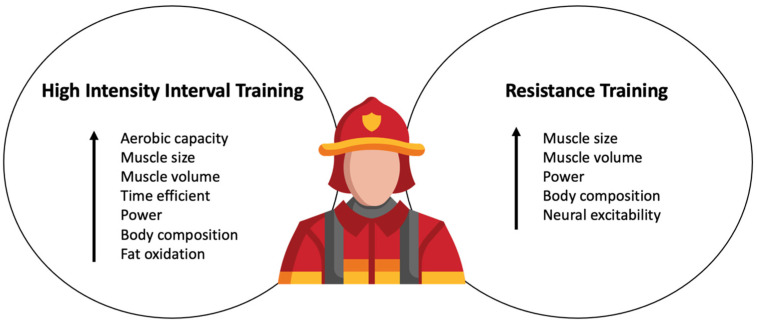
High-intensity interval training and resistance training offer many benefits to firefighters’ health and performance such as increased aerobic capacity, neuromuscular function, body composition, and metabolic adaptations.

**Figure 3 jfmk-09-00105-f003:**
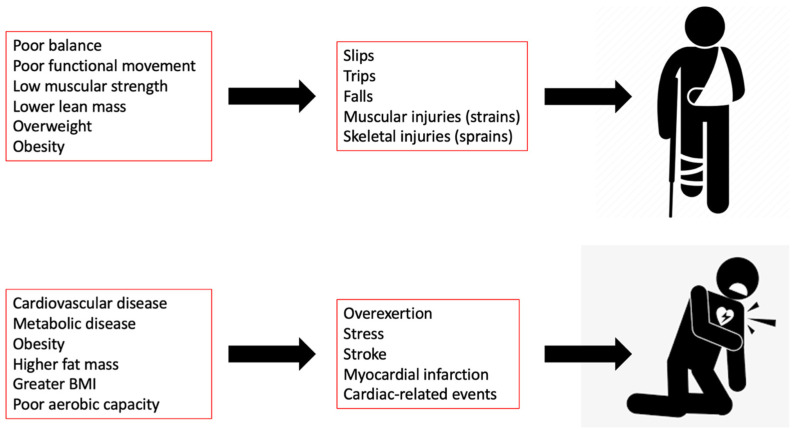
The literature shows several mechanistic links between health and performance variables and both non-fatal and fatal injuries in the fire service. Poor body composition and neuromuscular performance can impact functional movement on the job, leading to higher rates of musculoskeletal injuries. In addition, cardiometabolic disease and poor body composition are associated with higher levels of physiological stress and cardiac events in the fire service.

**Figure 4 jfmk-09-00105-f004:**
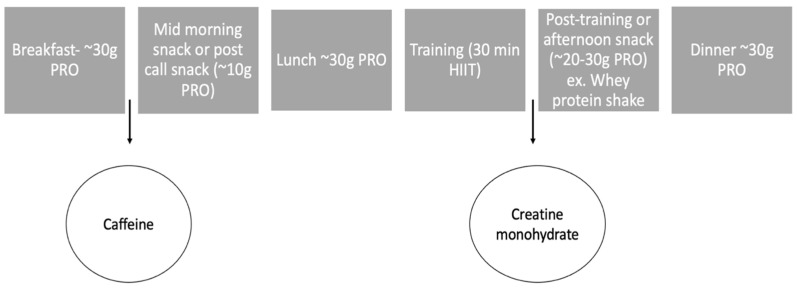
An example of implementing on-shift nutritional strategies and training for a 75 kg firefighter. Following the 1.4–2.0 g/kg protein recommendation, the range of protein intake should be 105–150 g/day. The assumption is the firefighter eating at normal mealtimes. It is suggested that caffeine is consumed in the morning to promote alertness, while consuming a snack mid-morning or post-call. Caffeine may help with grogginess as many firefighters note poor sleep quality. In the afternoon, the firefighter is partaking in 30 min of high-intensity interval training followed by protein consumption to help with satiety and recovery. To help meet training and body composition goals, creatine monohydrate should be added daily. Adequate protein consumption should help meet recovery needs based on training and call-volume as well as promote satiety throughout the shift.

**Figure 5 jfmk-09-00105-f005:**
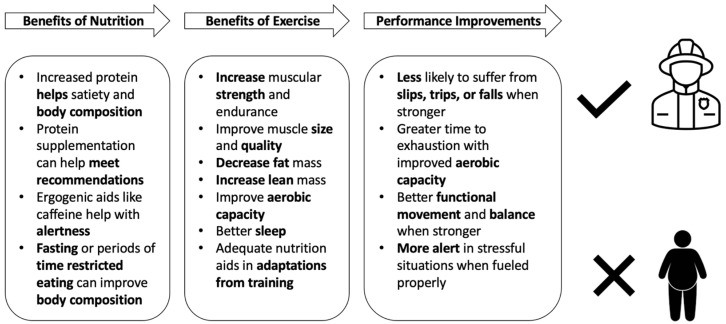
Adopting on-shift nutritional strategies such as increased protein intake, supplementation, addition of ergogenic aids, and caloric restriction can offer many benefits to overall firefighter health. Also, incorporating feasible exercise modalities such as high-intensity interval training and resistance training offer numerous advantages to firefighters. Taken together, improving nutrition and the adaptations from exercise can improve occupational performance in the fire service.

**Table 1 jfmk-09-00105-t001:** Dietary supplements and ergogenic aids that may be beneficial for firefighters to consume on-shift.

Supplement	Dose	Benefits	Magnitude of Effect
Whey Protein	20–30 g bolus *	Recovery	+++
		Muscle Protein Synthesis	
Essential Amino Acids	~10 g in whole foods or protein supplements *	Recovery	+
		Muscle Protein Synthesis	
Caffeine	3–6 milligrams per kilogram of body weight *	Alertness	++
		Improved Aerobic and Anaerobic Performance	
Creatine Monohydrate	Loading: 5 g, 4 times a day, 5–7 days = 20 g *	Neuroprotective	+++
		Cognitive Function	
	Maintenance: 3–5 g daily *	Anaerobic Performance	

* Adapted from the ISSN recommendations [45]. + indicates the magnitude of effect.

## Data Availability

No new data were created or analyzed in this study. Data sharing is not applicable to this article.

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
