# Peer review of "Strategies for Improving Firefighter Health On-Shift: A Review"

_jfmk, 2024, doi:10.3390/jfmk9020105_

Round 1
Reviewer 1 Report
Comments and Suggestions for Authors
This is a review paper that summarizes potential approaches that may improve health of firefighters, a group at high risks of diseases and injuries due to a demand and responsibilities. From their literature review, the authors suggested that improvements in both nutritional status and training may improve performance and health status of firefighters.
Although the manuscript is interesting and valuable to gain knowledge on benefits of nutrition and exercise on health status in general, the manuscript contains a number of issues to be clarified:
- It is difficult to understand health risks among firefighters. It is strongly suggested to re-structure the manuscript to first provide details about health problems of firefighters and then describe risk factors for these health problems in the introduction section. In addition, since the authors included implementation of suggested trainings and dietary options, it is suggested to describe current exercise and dietary practices of firefighters in the introduction.
- It is not clear why the authors focused on high intensity interval training and resistance trainings instead of other types of trainings, including endurance, aerobic training.
- Similarly, it is not certain why the authors focused on certain type of nutrients such as protein and encouraging energy restriction instead of other aspects of diet, such as appropriate energy intake and balanced meal. Aren’t eating at inappropriate timing or too much snacking, perhaps skipping breakfast risk factors for development of obesity or insufficient nutrient intakes? As a result, the authors’ suggestion about benefits of trainings and dietary options mentioned in the manuscript are not convincing.
- While the authors cited a number of studies, many were not conducted on firefighters and therefore there is a doubt that all references were not searched from the keywords stated in the methods section. It is suggested to state more details about methodology in the methods section.
- Furthermore, since the current manuscript contains the methods section, the authors should include the discussion section and conclusion of decent level of content.
Reviewer 2 Report
Comments and Suggestions for Authors
Considering the lack of scientific literature in this area, it is necessary to summarize what is currently known about firefighter health and how to implement on-duty strategies to improve well-being. I applaud the authors for doing an extensive search on firefighter literature, but also thinking "outside the box" to gain applicable insight from other tactical-based occupations. See specific comments below:
- Line 21: Not all fire departments have mandatory time for PA and eating which is why these two activities are incredibly difficult to plan on-shift. You describe this later on in this manuscript but I would edit this specific sentence to reflect that as well.
- Illustrations throughout the manuscript are helpful in summarizing "big picture" takeaways. Without the illustrations there is a LOT of information presented and it can be difficult to synthesize everything.
Overall, I have no major conflicts with this manuscript and only suggest very minor revisions. Well done!
Round 2
Reviewer 1 Report
Comments and Suggestions for Authors
I have completed a review of a revised manuscript titled "Strategies for improving firefighter health on-shift: a review (manuscript number: jfmk-3034692)". Thank you for incorporating my previous comments into the revised manuscript.
It would be better if the authors can provide further description on keywords they used to search literatures, including use of any combinations using "AND" and "OR" commands. However, if the editorial office cannot allow the authors for extra words count, then I have no further comments.
